# Upper Bound of Bayesian Generalization Error in Partial Concept Bottleneck Model

## Abstract

Concept Bottleneck Model (CBM) is a method for explaining neural networks. In CBM, concepts which correspond to reasons of outputs are inserted in the last intermediate layer as observed values. It is expected that we can interpret the relationship between the output and concept similar to linear regression. However, this interpretation requires observing all concepts and increases the generalization error of neural networks. Partial CBM (PCBM), which uses partially observed concepts, has been devised to resolve these difficulties. Although some numerical experiments suggest that the generalization error of PCBMs is almost as low as that of the original neural networks, the theoretical behavior of its generalization error has not been yet clarified because PCBM is singular statistical model. In this paper, we reveal the Bayesian generalization error in PCBM with a three-layered and linear architecture. The result indicates that the structure of partially observed concepts decreases the Bayesian generalization error compared with that of CBM (full-observed concepts).

## 1 Introduction

Methods of artificial intelligence such as neural networks have been widely applied in many research and practical areas (Goodfellow et al., 2016; Dong et al., 2021), increasing the demand for the interpretability of the model to deploy more intelligent systems to the real world. The accountability of such systems needs to be verified in fields related directly to human life, such as automobiles (self-driving systems (Xu et al., 2020)) and medicine (medical image analysis (Koh et al., 2020; Klimiene et al., 2022)). In these fields, there is an interest into models which are not black boxes, and therefore, various interpretable machine learning procedures have been investigated (Molnar, 2020). The concept bottleneck model (CBM) reported by Kumar et al. (2009); Lampert et al. (2009); Koh et al. (2020) is one of the architectures used to make the model interpretable. The CBM has a novel structure, called a concept bottleneck structure, wherein concepts are inserted between the output and last intermediate layers. In this structure, the last connection from the concepts to the output is linear and fully connected; thus, we can interpret the weights of that connection as the effect of the specified concept to the output, which is similar to the coefficients of linear regression. Concept-based interpretation is used in knowledge discovery for chess (McGrath et al., 2022), video representation (Qian et al., 2022), medical imaging (Hu et al., 2022), clinical risk prediction (Raghu et al., 2021), computer-aided diagnosis (Klimiene et al., 2022), and other healthcare domain problems (Chen et al., 2021). For this interpretation, concepts must be labeled accurately as explanations of inputs to predict outputs. For example, the concepts need to be set as clinical findings that are corrected by radiologists to predict the knee arthritis grades of patients based on X-ray images of their knee (Koh et al., 2020). In other words, CBM cannot be trained effectively without an accurate annotation from radiologists. Thus, the labeling cost is higher than that of the conventional supervised learning machine. Further, the concept bottleneck structure limits the parameter region of the network and it decreases the generalization error increases (Hayashi & Sawada, 2023).

Sawada & Nakamura (2022) proposed CBM with an additional unsupervised concept (CBM-AUC) to decrease the annotation cost of concepts. The core idea of CBM-AUC is that concepts are partially replaced as unsupervised values, and they are classified into tacit and explicit knowledge. The former concepts are provided as observations similar to that in the original CBM, i.e., they are supervised. The latter ones are not observable and obtained as output from the previous connection, i.e., they are unsupervised.

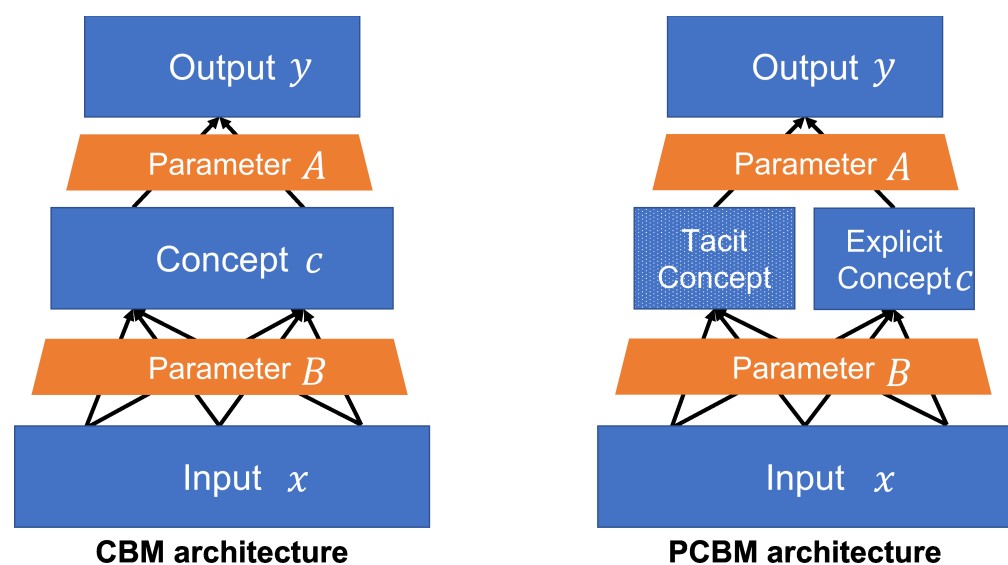

Figure 1: Schematics of CBM and PCBM architectures.

In the following, concepts corresponding to explicit/tacit knowledge are referred to as explicit/tacit concepts for simplicity. Futher, CBM-AUC uses a structure based on self-explaining neural networks (SENN) (Alvarez Melis & Jaakkola, 2018) for interpreting learned tacit concepts. In addition, partial CBM (PCBM) was developed in Li et al. (2022) and it only uses the above-mentioned core idea. For example, when the architecture is three-layered and linear (i.e., reduced rank regression), a neural network $y = ABx$ is trained, where $x$, $y$, and $(A, B)$ represent the input, output, and weight matrices, respectively. For CBM, there is an explicit concept vector $c$, and the weight parameters are learned as $y = ABx$ and $c = Bx$. The architectures of CBM and PCBM are illustrated in Figure 1. The detailed technical settings for learning: independent, sequential, and joint CBM (Koh et al., 2020), commonly represent the situation $y = ABx$ and $c = Bx$ in some forms. Alternatively, for PCBM and CBM-AUC, the dimension of the explicit concept vector is less than that of $Bx$. In other words, $Bx$ is partially supervised by explicit concepts as $c = B_2 x$, and the other part $B_1 x$ becomes the tacit concepts, where $B$ is vertically decomposed as $B = [B_1; B_2]$. Here, $B_1$ and $B_2$ are block matrices whose column dimensions are the same and the summation of their row dimensions is equal to the number of the rows in $B$. There are relevant variants of PCBM, which use different partitions for explicit and tacit concepts (Lu et al., 2021) and decoupling concepts (Zhang et al., 2022). The foundation of their structures is the network architecture of PCBM. Also, if the regularization term inspired by SENN in the loss function of CBM-AUC is zero, its loss function is equal to that of PCBM (Sawada & Nakamura, 2022; Li et al., 2022). Thus, in this paper, we consider only PCBM.

It has been shown empirically that PCBM outperforms the original CBM in terms of generalization (Li et al., 2022; Sawada & Nakamura, 2022); however, its theoretical generalization error has not yet been clarified. This is because neural networks are singular in general (Watanabe, 2007). Let $X^{(n)} = (X_1, \ldots, X_n)$ and $Y^{(n)} = (Y_1, \ldots, Y_n)$ be inputs and outputs of $n$ observations from $q(x, y) = q(y|x)q(x)$, respectively. Let $p(y|w, x)$ be a probability density function of a statistical model with a $d$-dimensional parameter $w$ and an input $x$, and $\varphi(w)$ represent a prior distribution. For instance, in the scenario wherein that a neural network is trained by minimizing a mean squared error, we set the model $p(y|w, x) \propto \exp(-\frac{1}{2}\|y - f(x; w)\|^2)$, where $f(x; w)$ is a neural network function parameterized by $w$. A statistical model is termed regular if the map from parameter $w$ to model $p$ is injective; otherwise, it is called singular (Watanabe, 2009; 2018). For neural networks, the map $w \mapsto f(\cdot; w)$ is not injective, i.e., there exists $(w_1, w_2)$ such that $f(x; w_1) = f(x; w_2)$ for any $x$. In the singular case, there are singularities in the zero point set of the Kullback-Leibler (KL) divergence between the data-generating distribution $q$ and $p$: $\{w \mid D_{\mathrm{KL}}(q\|p) = 0\}$. These singularities cause that a singular model has a hierarchical structure, i.e., candidate models are submodels. Because of

this structure, lower generalization error is obtained in the singular model compared to that of a regular model (Watanabe, 2000; 2001; 2009; 2018; Wei et al., 2022; Nagayasu & Watanbe, 2022). Let $G_n$ be the KL divergence between $q$ and the predictive distribution $p^*$: $G_n := D_{\mathrm{KL}}(q\|p^*)$. $G_n$ is called the Bayesian generalization error. If the model is regular, the expected $G_n$ is asymptotically dominated by half of the parameter dimension with an order of $1/n$: $\mathbb{E}[G_n] = d/2n + o(1/n)$; otherwise, there are a positive rational number $\lambda$ and an asymptotic behavior of $\mathbb{E}[G_n]$ as indicated below:

$$\mathbb{E}[G_n] = \frac{\lambda}{n} + o\left(\frac{1}{n}\right), \tag{1}$$

where $\lambda$ is called a real log canonical threshold (RLCT) (Watanabe, 2009; 2018). This theory is called the singular learning theory (Watanabe, 2009). The RLCTs of models depend on $(q, p, \varphi)$; thus, statisticians and machine learning researchers have analyzed them for each singular model. Furthermore, if the RLCT of the model is clarified, we can run effective sampling from the posterior distribution (Nagata & Watanabe, 2008) and select the optimal model (Drton & Plummer, 2017; Imai, 2019).

In previous research, the RLCT of CBM was clarified in the case with a three-layered and linear architecture network (Hayashi & Sawada, 2023). In this paper, we theoretically analyze RLCT, and based on the results, we derive an upper bound of the Bayesian generalization error in PCBM and prove it is less than that in CBM with assuming the same architecture. Note that this study aims to clarify the generalization error in PCBM and compare with that of CBM in order to evaluate the efficiency of the structure of PCBM since the structure of PCBM was provided for decreasing generalization error with a restriction that concepts were added to make the model interpretable. As a foundation for interpretable machine learning, it is important to elucidate the behavior of generalization errors in models with these constraints. In fact, there are reports suggesting that the determination of RLCTs contributes to interpretability (Hoogland et al., 2024; Anwar et al., 2024).

The remainder of this paper is organized as follows. In section 2, we introduce prior works that determine RLCTs of singular models and its application to statistics and machine learning. In section 3, we describe the framework of Bayesian inference when the data-generating distribution is not known, and we briefly explain the relationship between statistical models and RLCTs. In section 4, we state the main theorem. In section 5, we discuss our theoretical results from several perspectives, and in section 6, we conclude this paper. The proof of the main theorem is presented in appendix A.

## 2 Related Works

The RLCT depends on the triplet of the data-generating distribution, statistical model, and prior distribution, and therefore, we must consider resolution of singularities (Hironaka, 1964) for a family of functions on the real number field. In fact, there exist some procedures for resolving singularities for a single function on a algebraically closed field such as the complex number field (Hironaka, 1964). However, for the singular learning theory, a family of functions whose domain is a subset of the Euclidean space is considered. Currently, there is no standard method for calculating the theoretical value of the RLCT. That is why we need to identify the RLCT for each model.

Over the past two decades, RLCTs have been studied for various singular models. For example, mixture models, which are typical singular models (Hartigan, 1985; Watanabe, 2007), and their RLCTs have been analyzed for different types of component distributions: Gaussian (Yamazaki & Watanabe, 2003a), Bernoulli (Yamazaki & Kaji, 2013), Binomial (Yamazaki & Watanabe, 2004), Poisson (Sato & Watanabe, 2019), and etc. (Matsuda & Watanabe, 2003; Watanabe & Watanabe, 2022). Further, neural networks are also typical singular models (Fukumizu & Amari, 2000; Watanabe, 2001), and studies have been conducted to determine their RLCTs for cases where activation functions are linear (Aoyagi & Watanabe, 2005), analytic-odd (like tanh) (Watanabe, 2001), and Swish (Tanaka & Watanabe, 2020). Almost all learning machines are singular (Watanabe, 2007; Wei et al., 2022). Such instances of the singular learning theory applied for concrete models include the Boltzmann machines for several cases (Yamazaki & Watanabe, 2005b; Aoyagi, 2010a; 2013), matrix factorization with parameter restriction such as non-negative (Hayashi & Watanabe, 2017a;b; Hayashi, 2020) and simplex (equivalent to latent Dirichlet allocation) (Hayashi & Watanabe,

2020; Hayashi, 2021), latent class analysis (Drton, 2009), naive Bayes (Rusakov & Geiger, 2005), Bayesian networks (Yamazaki & Watanabe, 2003b), Markov models (Zwiernik, 2011), hidden Markov models (Yamazaki & Watanabe, 2005a), linear dynamical systems for prediction of a new series (Naito & Yamazaki, 2014), and Gaussian latent tree and forest models (Drton et al., 2017). We would like to emphasize that singular learning theory has a history of more than two decades, and as mentioned above, a lot of research has been conducted on RLCTs. Besides, recently, singular learning theory has been considered for investigating deep neural networks. Wei et al. (2022) reviewed the singular learning theory from the perspectives of deep learning. Aoyagi (2024) derived a deterministic algorithm for the deep linear neural network. Nagayasu and Watanabe clarified the asymptotic behavior of the Bayesian free energy in cases where the architecture is deep with ReLU activations (Nagayasu & Watanabe, 2023a) and convolutional with skip connections (Nagayasu & Watanabe, 2023b). Numerical verifications have also been conducted. Lau et al. (2023) proposed a scalable approximation of a localized version of the RLCT (a.k.a. local learning coefficient) using stochastic gradient Langevin dynamics (Welling & Teh, 2011). Experimental examples for modern scale networks to calculate local learning coefficient include superposition problems (Chen et al., 2023), deep linear neural networks (Furman & Lau, 2024), and transformers (Hoogland et al., 2024). In addition, these papers have experimentally demonstrated that RLCTs may contribute to the interpretability of neural networks.

From an application point of view, RLCTs are useful for performing Bayesian inference and solving model selection problems. Nagata & Watanabe (2008) proposed a procedure for designing exchange probabilities of inversed temperatures in the exchange Monte Carlo method. Imai (2019) derived an estimator of an RLCT and claimed that we can verify whether the numerical posterior distribution is precise by comparing the estimator and theoretical value. Drton & Plummer (2017) proposed a method called sBIC to select an appropriate model for knowledge discovery, which uses RLCTs of statistical models. Wei & Lau (2024) developed a variational method for Bayesian neural networks via resolution of singularities. Those studies are based on the framework of Bayesian inference.

## 3 Preliminaries

### 3.1 Framework of Bayesian Inference

Let $X^{(n)} = (X_1, \ldots, X_n)$ and $Y^{(n)} = (Y_1, \ldots, Y_n)$ represent collections of $n$ random variables. The function value of $(X_i, Y_i)$ is in $\mathcal{X} \times \mathcal{Y}$, where $\mathcal{X}$ and $\mathcal{Y}$ are subsets of finite-dimensional Euclidean or discrete spaces. In this article, the collections $X^{(n)}$ and $Y^{(n)}$ are referred to as the inputs and outputs, respectively. The pair $D_n := (X_i, Y_i)_{i=1}^n$ is called the dataset (a.k.a. sample) and its element $(X_i, Y_i)$ is called the ($i$-th) data. The sample is independently and identically distributed from the data-generating distribution (a.k.a. true distribution) $q(x, y) = q(y|x)q(x)$. From a mathematical point of view, the data-generating distribution is an induced probability measure of measurable functions $D_n$. Let $p(y|w, x)$ be a statistical model with a $d$-dimensional parameter $w \in \mathcal{W}$ and $\varphi(w)$ be a prior distribution, where $\mathcal{W} \subset \mathbb{R}^d$.

In Bayesian inference, we obtain the result of the parameter estimation as a distribution of the parameter, i.e., a posterior distribution. We define a posterior distribution as the distribution whose density is the function on $\mathcal{W}$, given as

$$\varphi^*(w|D_n) = \frac{1}{Z_n} \varphi(w) \prod_{i=1}^n p(Y_i|w, X_i), \tag{2}$$

where $Z_n$ is a normalizing constant used to satisfy the condition $\int \varphi^*(w|X^{(n)}) dw = 1$:

$$Z_n = \int dw \varphi(w) \prod_{i=1}^n p(Y_i|w, X_i). \tag{3}$$

$Z_n$ is called a marginal likelihood or partition function and its negative log value is called free energy $F_n := -\log Z_n$. We emphasize that $Z_n$ is also a probability distribution of $Y^{(n)}$ given $X^{(n)}$.

Let $\mathbb{E}[\cdot]$ be an expectation operator on overall datasets. The free energy appears as a leading term in the difference in log-likelihood between the data-generating distribution and model used for the dataset-generating process. In other words, as a function of models, $D_{\mathrm{KL}}(Q\|Z_n)$ only depends on $\mathbb{E}[F_n]$, where

$Q(Y^{(n)}|X^{(n)}) = \prod_{i=1}^{n} q(Y_i|X_i)$ is the distribution generating the output dataset given the input dataset. For the model-selection problem, the marginal likelihood leads to a model maximizing a posterior distribution of model size (such as the number of hidden units of neural networks). This perspective is called knowledge discovery.

Evaluating the dissimilarity between the true and the predicted value is also important for statistics and machine learning. This perspective is called prediction. A predictive distribution is defined by the following density function of an output $y \in \mathcal{Y}$ with a new input $x \in \mathcal{X}$.

$$p^*(y|D_n, x) = \int dw \varphi^*(w|D_n) p(y|w, x). \tag{4}$$

A Bayesian generalization error $G_n$ is defined by the KL divergence between the data-generating distribution and predictive one, given as

$$G_n = \iint dx dy q(x) q(y|x) \log \frac{q(y|x)}{p^*(y|D_n, x)}. \tag{5}$$

Obviously, it is the dissimilarity between the true and predictive distribution in terms of KL divergence.

Both of these perspectives consider the scenario wherein the data is random variable and its source $q(y|x)$ is unknown. This situation is considered generic in the real world data analysis (McElreath, 2020; Watanabe, 2023). Moreover, in general, the model is singular when it has a hierarchical structure or latent variables (Watanabe, 2009; 2018), such as models written in section 2.

### 3.2 Singular Learning Theory

We briefly intoduce some important properties of the singular learning theory. First, several concepts are defined. Let $S$ and $S_n$ be

$$S = -\int dx dy q(x) q(y|x) \log q(y|x), \tag{6}$$

$$S_n = -\frac{1}{n} \sum_{i=1}^{n} \log q(Y_i|X_i). \tag{7}$$

$S$ and $S_n$ are called the entropy and empirical entropy, respectively. The KL divergence between the data-generating distribution and statistical model is denoted by

$$K(w) = \int dy dx q(x, y) \log \frac{q(y|x)}{p(y|w, x)} \tag{8}$$

as a non-negative function of parameter $w$. This is called an averaged error function based on Watanabe (2018).

As technical assumptions, we suppose the parameter set $\mathcal{W} \subset \mathbb{R}^d$ is compact, $K^{-1}(0)$ is not empty, and the prior is positive and bounded on $K^{-1}(0)$, i.e., $0 < \varphi(w) < \infty$ holds for any $w \in K^{-1}(0)$, where

$$K^{-1}(0) := \{w \in \mathcal{W} \mid K(w) = 0\}. \tag{9}$$

In addition, we assume that $\varphi(w)$ is a $C^\infty$-function on $\mathcal{W}$ and $K(w)$ is an analytic function on $\mathcal{W}$. For the sake of simplicity, we assume $K^{-1}(0)$ is not empty: the realizable case. In fact, if the true distribution cannot be realized by the model candidates, we can redefine the averaged error function as $D_{\mathrm{KL}}(p^0\|p)$: the KL divergence between the nearest model to the data-generating distribution and candidate model, where $w^0 = \arg\min D_{\mathrm{KL}}(q\|p)$ and $p^0(y|x) = p(y|w^0, x)$, and therefore, we can expand the singular learning theory for the non-realizable cases (Watanabe, 2010; 2018).

The RLCT of the model is defined by the following. Let $\Re(z)$ be the real part of a complex number $z$.

**Definition 3.1** (RLCT)**.** *Let $z \mapsto \zeta(z)$ be the following univariate complex function,*

$$\zeta(z) = \int dw \varphi(w) K(w)^z. \tag{10}$$

$\zeta(z)$ *is holomorphic on* $\Re(z) > 0$. *Further, it can be analytically continued on the entire complex plane as a meromorphic funcion. Its poles are negative rational numbers. The maximum pole is denoted by* $(-\lambda)$, *and* $\lambda$ *is the RLCT of the model with regard to* $K(w)$.

Mathematical properties and details of the relation between the above type zeta function and resolution of singularities is described in Atiyah (1970); Bernstein (1972); Sato & Shintani (1974). We refer to the above complex function $\zeta(z)$ as the zeta function of learning theory. In general, the RLCT is determined by the triplet that consists of the true distribution, the model, and the prior: $(q, p, \varphi)$. If the prior becomes zero or infinity on $K^{-1}(0)$, it affects the RLCT; otherwise, the RLCT is not affected by the prior and becomes the maximum pole of the following zeta function of learning theory.

$$\zeta(z) = \int dw K(w)^z. \tag{11}$$

We refer to the RLCT determined by the maximum pole of the above as the RLCT with regard to $K(w)$.

When the model is regular, $K^{-1}(0) = \{w^0\}$ is a point in the parameter space, and we can expand $K(w)$ around $w^0$ as

$$K(w) = (w - w^0)^T H(w^*)(w - w^0), \tag{12}$$

where $w^*$ exists in the neighborhood of $w^0$ and $H(w)$ is the Hessian matrix of $K(w)$. Note that $K(w^0) = \nabla K(w^0) = 0$ holds in the regular case. It is a quadratic form and there is a diffeomorphism $w = f(u)$ such that

$$K(f(u)) = u_1^2 + \ldots + u_d^2. \tag{13}$$

By using this representation, we immediately obtain its RLCT from the definition, i.e., the RLCT is equal to $d/2$. However, in general, the averaged error function cannot be expanded as a quadratic form since $K^{-1}(0)$ is not a point. If the prior satisfies $0 < \varphi(w) < \infty$ for any $w \in K^{-1}(0)$, $d/2$ is an upper bound of the RLCT and its tightness is often vacuous. Watanabe had resolved this issue by using resolution of singularities (Hironaka, 1964) for $K^{-1}(0)$ (Watanabe, 2001; 2010). Thus, we have the following form even if the model is singular. According to resolution of singularities theorem (Hironaka, 1964; Atiyah, 1970), there is a manifold $\mathcal{M}$ and birational map $g : \mathcal{M} \to \mathcal{W}$ such that

$$K(g(u)) = u_1^{2k_1} \ldots u_d^{2k_d}, \tag{14}$$

$$|g'(u)| = u_1^{h_1} \ldots u_d^{h_d}. \tag{15}$$

This is called the normal crossing form.

As a demonstration, we describe how to calculate an RLCT from resolution of singularities. This explanation is based on Watanabe (2018). When we construct the normal crossing form of the averaged error functon for a singular model, we can calculate its RLCT as follows. Using the normal crossing form and variable transformation, we consider the following zeta function to calculate the RLCT.

$$\zeta(z) = \int du u_1^{2k_1 z} \ldots u_d^{2k_d z} u_1^{h_1} \ldots u_d^{h_d} \tag{16}$$

$$= \int du u_1^{2k_1 z + u_1^{h_1}} \ldots u_d^{2k_d z + u_d^{h_d}}. \tag{17}$$

Since the parameter set is compact, considering a partition of unity $\sum_a \phi_a(u)$ for $\mathcal{M}$, we have

$$\zeta(z) = \sum_a \int du u_1^{2k_1 z + u_1^{h_1}} \ldots u_d^{2k_d z + u_d^{h_d}} \phi_a(u) \tag{18}$$

$$= \sum_a \int_{[0,1]^d} du u_1^{2k_1 z + h_1} \ldots u_d^{2k_d z + h_d}. \tag{19}$$

For each local coordinate whose index is $a$, we have

$$\int_{[0,1]^d} du u_1^{2k_1 z + h_1} \ldots u_d^{2k_d z + h_d} = \prod_{j=1}^d \int_0^1 du_j u_j^{2k_j z + h_j} = \prod_{j=1}^d \frac{1}{2k_j z + h_j + 1}. \tag{20}$$

The zeta function is meromorphic because of analytic continuation. Thus, the negative maximum pole, i.e., the RLCT $\lambda$ is equal to

$$\lambda = \min_{a} \min_{j=1}^{d} \frac{h_j + 1}{2k_j}. \tag{21}$$

Therefore, when we find a map for resolution of singularities, we can determine the RLCT. As mentioned in section 2, there is no standard method to construct the map and many researches have been carried out to calculate RLCTs.

The asymptotic behaviors of the free energy and the Bayesian generalization error have been proved.

**Theorem 3.1.** *Let $\lambda$ be the RLCT with regard to $K(w)$. The free energy $F_n$ and the Bayesian generalization error $G_n$ satisfies*

$$F_n = nS_n + \lambda \log n + O_p(\log \log n), \tag{22}$$

$$\mathbb{E}[G_n] = \frac{\lambda}{n} + o\left(\frac{1}{n}\right). \tag{23}$$

The proof of the above facts and the details of the singular learning theory are described in Watanabe (2009) and Watanabe (2018). As mentioned in section 2, there is no standard method for deterministic construction of $\mathcal{M}$ and $g$; thus, prior works have found them or resolved relaxed cases to derive an upper bound of the RLCTs. For example, Aoyagi & Watanabe (2005) clarified the exact value of the RLCT for the three-layered and linear neural network by constructing resolution of singularity.

**Definition 3.2** (RLCT of Reduced Rank Regression). *Consider a three-layered and linear neural network (a.k.a. reduced rank regression) with $N$-dimensonal input, $H$-dimensional intermediate layer, and $M$-dimensional output. Let $U$ and $V$ be real matrices whose sizes are $M \times H$ and $H \times N$, respectively; they are the weight matrices of the model. The true parameters with regard to $U$ and $V$ are denoted by $U^0$ and $V^0$, respectively. A zeta function of learning theory is defined as follows:*

$$\zeta_{\mathrm{R}}(z) = \iint dU \, dV \, \|UV - U^0 V^0\|^{2z}. \tag{24}$$

*It is holomorphic in $\Re(z) > 0$ and can be analytically continued as a meromorphic function on the entire complex plain. The maximum pole of $\zeta_{\mathrm{R}}(z)$ is denoted by $(-\lambda_{\mathrm{R}})$. Then, $\lambda_{\mathrm{R}}$ is called the RLCT of three-layered and linear neural network.*

The following theorem clarifies the RLCT of reduced rank regression.

**Theorem 3.2** (Aoyagi & Watanabe (2005)). *Let $r$ be the true rank: $r = \mathrm{rank}(U^0 V^0)$. The RLCT $\lambda_{\mathrm{R}}$ is obtained as follows:*

1. *If $M + r \leqq N + H$ and $N + r \leqq M + H$ and $H + r \leqq N + M$*

   *(a) and $N + M + H + r$ is even,*

   $$\lambda_{\mathrm{R}} = \frac{1}{8}\{2(H + r)(N + M) - (N - M)^2 - (H + r)^2\}. \tag{25}$$

   *(b) and $N + M + H + r$ is odd,*

   $$\lambda_{\mathrm{R}} = \frac{1}{8}\{2(H + r)(N + M) - (N - M)^2 - (H + r)^2 + 1\}. \tag{26}$$

2. *If $N + H < M + r$,*

   $$\lambda_{\mathrm{R}} = \frac{1}{2}\{HN + r(M - H)\}. \tag{27}$$

3. *If $M + H < N + r$,*

$$\lambda_{\mathrm{R}} = \frac{1}{2}\{HM + r(N - H)\}. \tag{28}$$

4. *Otherwise, i.e. if $N + M < H + r$,*

$$\lambda_{\mathrm{R}} = \frac{1}{2}N(M + K). \tag{29}$$

We aim to transform $K(w)$ into a normal crossing form and relax $K(w)$ to derive an upper bound of the RLCT of PCBM.

## 4 Main Theorem

Let $N$, $H$, and $M$ be the dimensions of the output, hidden layer, and input, respectively. The hidden layer is decomposed by $H_1$-dimensional learnable units and $H_2$-dimensional observable concepts, and $H = H_1 + H_2$ holds. The true dimension of the learnable units is denoted by $H_1^0$.

Let $x \in \mathbb{R}^N$, $c \in \mathbb{R}^{H_2}$, and $y \in \mathbb{R}^M$, respectively. Define $A$ and $B$ as real matrices whose sizes are $M \times H$ and $H \times N$. We consider the block matrices of $A$ and $B$. $A_1$, $A_2$, $B_1$, and $B_2$ denote matrices whose sizes are $M \times H_1$, $M \times H_2$, $H_1 \times N$, and $H_2 \times N$, respectively. Assume that $A$ is horizontally concatenated by $A_1$ and $A_2$ and $B$ is vertically concatenated by $B_1$ and $B_2$; $A = [A_1, A_2]$ and $B = [B_1; B_2]$. Similarly, by replacing $H_1$ to $H_1^0$, we define matrices and their block-decomposed representation as $A^0 := [A_1^0, A_2^0]$ and $B^0 := [B_1^0; B_2^0]$. They are the true parameters corresponding to $A = [A_1, A_2]$ and $B = [B_1; B_2]$, respectively. In the following, the input $x$ is observable, $w = (A, B)$ is a parameter and the output $y$ and concept $c$ is randomly generated by the data-generating distribution conditioned by $x$. $\|\cdot\|$ of a matrix is denoted by a Frobenius norm.

Here, along with Definition 3.1, we define the RLCT of PCBM as follows.

**Definition 4.1** (RLCT of PCBM). *Let $(-\lambda_{\mathrm{P}})$ be the maximum pole of the following complex function $z \mapsto \zeta(z)$,*

$$\zeta(z) = \iint dA dB (\|AB - A^0 B^0\|^2 + \|B_2 - B_2^0\|^2)^z, \tag{30}$$

*where $\zeta(z)$ is holomorphic on $\Re(z) > 0$ and can be analytically continued on the entire complex plane as a meromorphic funcion. Then, $\lambda_{\mathrm{P}}$ represents the RLCT of PCBM.*

It is immediately derived that $\lambda_{\mathrm{P}}$ is a positive rational number. In this article, we prove the following theorem.

**Theorem 4.1** (Main Theorem). *The RLCT of PCBM $\lambda_{\mathrm{P}}$ satisfies the following inequality:*

$$\lambda_{\mathrm{P}} \leqq \lambda_{\mathrm{R}}(M, H_1, N, \mathrm{rank}(A_1^0 B_1^0)) + \frac{H_2(M + N)}{2}, \tag{31}$$

*where $\lambda_{\mathrm{R}}(N, H_1, M, r)$ is the RLCT of reduced rank regression in Theorem 3.2 when the dimensions of the inputs, hidden layer, and outputs are $N$, $H_1$, and $M$, respectively, and the true rank is $r$.*

We prove the above theorem in appendix A. As an application of the main theorem, we derive an upper bound of the Bayesian generalization error in PCBM.

**Theorem 4.2** (Bayesian Generalization Error in PCBM). *We define the probability distributions of $(y, c)$ conditioned by $x$*

$$q(y, c|x) \propto \exp\left(-\frac{1}{2}\|y - A^0 B^0 x\|^2\right) \exp\left(-\frac{1}{2}\|c - B^0 x\|^2\right), \tag{32}$$

$$p(y, c|A, B, x) \propto \exp\left(-\frac{1}{2}\|y - ABx\|^2\right) \exp\left(-\frac{1}{2}\|c - Bx\|^2\right). \tag{33}$$

*Further, let $\varphi(A,B)$ be a prior distribution whose density is positive and bounded on $K(A,B) = 0$, where $K(A,B) = D_{\mathrm{KL}}(q\|p)$. Then, the expected generalization error $\mathbb{E}[G_n]$ asymptotically has the following upper bound:*

$$\mathbb{E}[G_n] \leqq \frac{1}{n}\left(\lambda_{\mathrm{R}}(M, H_1, N, \mathrm{rank}(A_1^0 B_1^0)) + \frac{H_2(M+N)}{2}\right) + o\left(\frac{1}{n}\right). \tag{34}$$

If Theorem 4.1 is proved, Theorem 4.2 is immediately obtained. Therefore, we set Theorem 4.1 as the main theorem of this paper.

## 5 Discussion

We discuss the main result of this paper from six points of view as well as the remaining issues.

**Theoretical Free Energy**   First, we focus the other criterion of Bayesian inference: the marginal likelihood. In this paper, we analyzed the RLCT of PCBM with a three-layered and linear architecture, which resulted in obtaining Theorem 4.2; the theoretical behavior of the Bayesian generalization error is clarified. In addition, we derive the upper bound of the free energy $F_n$. According to Theorem 3.1, we have

$$F_n = nS_n + \lambda_{\mathrm{P}} \log n + O_p(\log \log n), \tag{35}$$

where $\lambda_{\mathrm{P}}$ is the RLCT in Theorem 4.1 and $S_n$ is the empirical entropy. Thus, we have the following inequality: an upper bound of the free energy in PCBM.

$$F_n - nS_n \leqq \left[\lambda_{\mathrm{R}}(M, H_1, N, \mathrm{rank}(A_1^0 B_1^0)) + \frac{H_2(M+N)}{2}\right] \log n + O_p(\log \log n). \tag{36}$$

There exists an information criterion that uses RLCTs: sBIC (Drton & Plummer, 2017). Further, the non-trivial upper bound of an RLCT is useful for approximating the free energy by sBIC (Drton & Plummer, 2017; Drton et al., 2017). PCBM is an interpretable machine learning model; thus, it can be applied to not only prediction of unknown data but also explanation of phenomenon, i.e., knowledge discovery. Evaluation based on marginal likelihood is conducted in knowledge discovery (Good et al., 1966; Schwarz, 1978). Hence, our result also contributes resolving the model-selection problems of PCBM.

**Bayesian predictive distribution and stochastic gradient descent**   Next, we consider that there is a potential expansion of our main result. The Bayesian generalization error depends on the model with regard to the predictive distribution:

$$p^*(y,c|x) = \iint dA dB\, p(y,c|A,B,x)\varphi^*(A,B|Y^{(n)}, C^{(n)}, X^{(n)}), \tag{37}$$

where $Y^{(n)}$, $C^{(n)}$, and $X^{(n)}$ are the dataset of the output, concept, and input, respectively. If the posterior distribution was a delta distribution whose mass was on an estimator $(\hat{A}, \hat{B})$, the predictive one would be the model whose parameter is the estimator:

$$p^*(y,c|x) = \iint dA dB\, p(y,c|A,B,x)\delta(\hat{A},\hat{B}) \tag{38}$$

$$= p(y,c|\hat{A},\hat{B},x). \tag{39}$$

Thus, the Bayesian predictive distribution namely includes point estimation. Recently, many neural networks are trained by parameter optimization with mini-batch stochastic gradient descent (SGD). We believe that it is an important issue what is the difference between the generalization errors of the Bayesian and other point estimations. There are many theoretical facts that the Bayesian posterior distribution dominates the stationary distribution of the parameter optimized by the mini-batch SGD (Şimşekli, 2017; Mandt et al., 2017; Smith et al., 2018). Besides, Furman and Lau empirically demonstrated that RLCT measures the

model capacity and complexity of neural network for the modern scale at least in the case where the activations are linear (Furman & Lau, 2024) effectively via stochastic gradient Langevin dynamics (Welling & Teh, 2011; Lau et al., 2023). Hence, the method based on singular learning theory, which analyzes the Bayesian generalization error through RLCTs, has a probability of contributing to the generalization error evaluation of learning by the mini-batch SGD.

**Potential Application to Transfer Learning** Although we treat a three-layered and linear neural network, we consider a potential application to transfer learning. There is a method for constructing features from a state-of-the-art deep neural network, including vectors in some middle layers in the context of transfer learning (Yosinski et al., 2014). Here, the original input is transformed to the feature vector through the frozen deep network. Using these features as inputs of three-layered and linear PCBM and learning it, our main result can be applied for its Bayesian generalization error, which corresponds to connecting the PCBM to the last layer of the frozen deep network and learning weights in the PCBM part, where we consider transferring the trained and frozen network to other domains and adding interpretability using concepts. In practice, the efficiency and accuracy of such a method needs to be evaluated by numerical experiments; however, we only show the above potential application because this work aims at the theoretical analysis of the Bayesian generalization error.

**Effect of Structure of PCBM** The main theorem suggests that PCBM should outperform CBM. In the previous research (Hayashi & Sawada, 2023), the exact RLCT of CBM $\lambda_C$ is clarified for the three-layered and linear CBM: $\lambda_C = \frac{H(M+N)}{2}$, where $H$ is the number of units in the intermediate layer and equal to the dimension of the concept. Besides, since $\lambda_R(M, H_1, N, r)$ is the RLCT of the three-layered and linear neural network, its trivial upper bound is $\frac{H_1(M+N)}{2}$: a half of the parameter dimension. Therefore,

$$\lambda_P \leqq \lambda_R(M, H_1, N, r') + \frac{H_2(M+N)}{2} \tag{40}$$

$$\leqq \frac{H_1(M+N)}{2} + \frac{H_2(M+N)}{2} \tag{41}$$

$$= \frac{H(M+N)}{2} \tag{42}$$

$$= \lambda_C \tag{43}$$

holds, where $r' = \text{rank}(A_1^0 B_1^0)$ in Theorem 4.1. Let $G_C$ and $G_P$ be the expected Bayesian generalization error in CBM and PCBM, respectively. Then, we have

$$\begin{cases} G_P = \frac{\lambda_P}{n} + o\left(\frac{1}{n}\right), \\ G_C = \frac{\lambda_C}{n} + o\left(\frac{1}{n}\right). \end{cases} \tag{44}$$

Because $\lambda_P \leqq \lambda_C$,

$$G_P \leqq G_C + o\left(\frac{1}{n}\right). \tag{45}$$

Therefore, the Bayesian generalization error in PCBM is less than that in CBM. To consider the effect of PCBM, i.e., partial replacing concepts from supervised to unsupervised, we revisit the following decomposition of the RLCT of CBM,

$$\lambda_C = \frac{H_1(M+N)}{2} + \frac{H_2(M+N)}{2}. \tag{46}$$

By replacing the first term, the right-hand side becomes an upper bound in Theorem 4.1 and it is less than the left-hand side. Since this replacement makes the $H_1$-dimensional concept unsupervised (a.k.a. tacit) in CBM, i.e., constructing PCBM, the structure of PCBM improves the generalization error compared to that of CBM. This is because the supervised (a.k.a. explicit) concepts are partially given in the middle layer of PCBM. In addition, we can find a lower bound of the deference of the generalization error $\lambda_C/n - \lambda_P/n$. The following corollary is immediately proved because of the above discussion and Theorems 4.1 and 4.2.

**Corollary 5.1** (Lower Bound of Generalization Error Difference between CBM and PCBM)**.** *In a three-layered neural network with $N$-dimensional input, $H$-dimensional middle layer, and $M$-dimensional output, the expected Bayesian generalization error is at least*

$$G_{\mathrm{C}} - G_{\mathrm{P}} \geqq \frac{1}{n} \left[ \frac{H_1(M+N)}{2} - \lambda_{\mathrm{R}}(M, H_1, N, r') \right] + o\left(\frac{1}{n}\right) \tag{47}$$

*smaller for PCBM, which gives the observations for only the $H_2$ dimension of the middle layer, than for CBM, which gives the observations for all of the middle layers, where $H_1 = H - H_2$.*

Note that the dimensions of observation in PCBM and CBM are different because the numbers of supervised concepts in them do not eqaul. We can use this corollary to decrease the concept dimension when we plan the dataset collection. This is just a result for shallow networks; however, it contributes the foundation to clarify the effect of the concept bottleneck structure for model prediction performance. Indeed, some experimental examinations demonstrate that PCBM and its variant (such as CBM-AUC) outperform CBM (Sawada & Nakamura, 2022; Li et al., 2022; Lu et al., 2021; Zhang et al., 2022).

**Suggestion for Efficiency of Simultaneous Learning Parameter**   In the above paragraph, we showed a perspective of the network structure for the upper bound in Theorem 4.1. There exists another point of view: a direct interpretation of it. In PCBM, we train both the part of tacit concepts and that of explicit ones, simultaneously. According to the proof of the main theorem in the appendix A, our upper bound is the RLCT with regard to the averaged error function

$$\overline{K}(A, B) = \|A_1 B_1 - A_1^0 B_1^0\|^2 + \|A_2 B_2 - A_2^0 B_2^0\|^2 + \|B_2 - B_2^0\|^2. \tag{48}$$

We can refer it to the following model called the upper model. This model separately learns the part of tacit concepts and that of explicit ones. The former is a neural network whose averaged error function is $\|A_1 B_1 - A_1^0 B_1^0\|^2$: a three-layered and linear neural network, and the latter is another neural network whose averaged error function is $\|A_2 B_2 - A_2^0 B_2^0\|^2 + \|B_2 - B_2^0\|^2$: a CBM. In the upper model, the former and latter are independent since there is no intersection of parameters, i.e., the former only depends on $(A_1, B_1)$ and the latter on $(A_2, B_2)$. Hence, our main result shows that PCBM is preferred to the upper model for generalization. The upper model is just artificial; however, the inequality of Theorem 4.1 suggests that simultaneous training is better than separately training (multi-stage estimation) such as independent and sequential CBM (Koh et al., 2020). Indeed, similar phenomenon have been observed in constructing graphical models (Sawada & Hontani, 2012; Hontani et al., 2013), representation learning (Collobert et al., 2011; Krizhevsky et al., 2012), and pose estimation (Tobeta et al., 2022).

**Case of Categorical Data**   We consider the data type of the outputs and concepts. The main theorem assumes that the objective variable (output) and concept are real vectors. However, categorical varibales can be outputs and concepts like wing color in a bird species classification task (Koh et al., 2020). The prior study concerning the RLCT of CBM (Hayashi & Sawada, 2023) clarified the asymptotic Bayesian generalization error in CBM when not only the output and concept are real but also when at least one of them is categorical. According to this result, we derive how the upper bound of the RLCT in the main theorem behaves if the data type changes. The upper bound has two terms: the RLCT of reduced rank regression for the tacit concept and that of CBM for the explicit concept. Let $\overline{\lambda}_1$ and $\overline{\lambda}_2$ be the first and second term of the upper bound in Theorem 4.1:

$$\begin{cases} \overline{\lambda}_1 = \lambda_{\mathrm{R}}(M, H_1, N, r'), \\ \overline{\lambda}_2 = \frac{H_2(M+N)}{2}. \end{cases} \tag{49}$$

$\overline{\lambda}_1$ depends on only the type of the output since all concepts in this part are unsupervised. On the other hand, $\overline{\lambda}_2$ is determined by the type of the concept as well as that of the output. From the probability distribution point of view, the source distribution is replaced from Gaussian to categorical in Theorem 4.2 if categorical variables are generated. Hence, we should also replace the zeta function defining the RLCT in Definition 4.1 to the appropriate form for the KL divergence between categorical distributions. Meanwhile,

concepts can be binary vectors such as birds' features e.g., whether the wing is black or not. In this case, the concept part of the data-generating distribution is replaced from Gaussian to Bernoulli. By using Corollaries 4.1 and 4.2 in (Hayashi & Sawada, 2023), we immediately obtain the following result.

**Corollary 5.2.** *Let $H_2^{\mathrm{r}}$ and $H_2^{\mathrm{c}}$ be the dimension of the real and categorical concept, respectively. The dimension of the real and categorical output are denoted by $M^{\mathrm{r}}$ and $M^{\mathrm{c}}$. Then, we have*

$$\lambda_1 = \lambda_{\mathrm{R}}(M^{\mathrm{r}} + M^{\mathrm{c}} - 1, H_1, N, r'), \tag{50}$$

$$\lambda_2 = \frac{1}{2}(H_2^{\mathrm{r}} + H_2^{\mathrm{c}})(M^{\mathrm{r}} + M^{\mathrm{c}} + N - 1), \tag{51}$$

*i.e., the following holds:*

$$\lambda_{\mathrm{P}} \leqq \lambda_{\mathrm{R}}(M^{\mathrm{r}} + M^{\mathrm{c}} - 1, H_1, N, r') + \frac{1}{2}(H_2^{\mathrm{r}} + H_2^{\mathrm{c}})(M^{\mathrm{r}} + M^{\mathrm{c}} + N - 1). \tag{52}$$

Therefore, we can expand our main result to categorical data.

**Remaining Problems** Lastly, remaining problems are discussed. As mentioned above, it is important to clarify the difference between the generalization error of Bayesian inference and that of optimization by mini-batch SGD. This research aims to perform the theoretical analysis of the RLCT; thus, numerical behaviors have not been demonstrated. The other issues are as follows. Our result is suitable for three-layered and linear architectures. For the shallowness, the RLCT of the deep linear neural network has been clarified in (Aoyagi, 2024). For the linearity, some non-linear activations are studied for usual neural networks in the case of three-layered architectures (Watanabe, 2001; Tanaka & Watanabe, 2020). Besides, Vandermonde-matrix-type singularities have been analyzed to establish a multi-purpose resolution method (Aoyagi, 2010b; 2019). However, these prior results are not for PCBM. It is non-trivial whether these works can be applied to deep and non-linear PCBM. In addition, when the activations are non-linear, the structure of $K^{-1}(0)$ and its singularities become complicated even if the network is shallow. Therefore, there are challenging problems for the shallowness and linearity. The other issue is clarifying the theoretical generalization error of PCBM variants and how different it is from that of PCBM. There are other PCBM variants such as CBM-AUC (Sawada & Nakamura, 2022), explicit and implicit coupling (Lu et al., 2021), and decoupling (Zhang et al., 2022). They have various structures. For example, in CBM-AUC, they add a regularization term that makes concepts more interpretable using SENN to the loss function. If the main loss is based on the likelihood, the regularization term is referred to the prior. However, SENN has some derivative restrictions as a regularization term and it is non-trivial to find a distribution corresponding to the restriction, making it difficult to ascribe the generalization error analysis of CBM-AUC to the singular learning theory.

## 6 Conclusion

In this paper, we mathematically derived an upper bound of the real log canonical threshold (RLCT) for partical concept bottleneck model (PCBM) and a theoretical upper bound of the Bayesian generalization error and the free energy. Further, we showed that PCBM outperforms the conventional concept bottleneck model (CBM) in terms of generalization and provided a lower bound of the Bayesian generalization error difference between CBM and PCBM.

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

# A    Proof of Main Theorem

*Proof.* According to Definition 4.1, the RLCT of PCBM is determined by the zero points $K^{-1}(0)$ of the averaged error function

$$K(A, B) = \|AB - A^0 B^0\|^2 + \|B_2 - B_2^0\|^2. \tag{53}$$

Thus, considering order isomorphism of RLCTs, we can derive an upper bound of the RLCT by evaluating the averaged error function.

Decomposing matrices, we have

$$\|AB - A^0 B^0\|^2 \tag{54}$$

$$= \|[A_1, A_2][B_1; B_2] - [A_1^0, A_2^0][B_1^0; B_2^0]\|^2 \tag{55}$$

$$= \|A_1 B_1 + A_2 B_2 - (A_1^0 B_1^0 + A_2^0 B_2^0)\|^2. \tag{56}$$

By using the triangle inequality, we obtain

$$\|AB - A^0 B^0\|^2 + \|B_2 - B_2^0\|^2 \tag{57}$$

$$\leqq \|A_1 B_1 - A_1^0 B_1^0\|^2 + \|A_2 B_2 - A_2^0 B_2^0\|^2 + \|B_2 - B_2^0\|^2. \tag{58}$$

Let $\overline{K}(A, B)$ be the right-hand side of the above. Considering $\overline{K}(A, B) = 0$, we have the following joint equation:

$$\begin{cases} \|A_1 B_1 - A_1^0 B_1^0\|^2 = 0, \\ \|A_2 B_2 - A_2^0 B_2^0\|^2 = 0, \\ \|B_2 - B_2^0\|^2 = 0. \end{cases} \tag{59}$$

Using the third $B_2 = B_2^0$, we solve the second equation and

$$\begin{cases} \|A_1 B_1 - A_1^0 B_1^0\|^2 = 0, \\ \|A_2 - A_2^0\|^2 = 0, \\ \|B_2 - B_2^0\|^2 = 0 \end{cases} \tag{60}$$

holds. Let $\overline{\lambda}_1$ and $\overline{\lambda}_2$ be the RLCT with regard to $\|A_1 B_1 - A_1^0 B_1^0\|^2$ and $\|A_2 - A_2^0\|^2 + \|B_2 - B_2^0\|^2$, respectively. Since the parameter in the first equation $(A_1, B_1)$ and that in the second and third $(A_2, B_2)$ are independent, the RLCT with regard to $\overline{K}(A, B)$ becomes the sum of $\overline{\lambda}_1$ and $\overline{\lambda}_2$. For $\overline{\lambda}_1$, by using Theorem 3.2, we have

$$\overline{\lambda}_1 = \lambda_{\mathrm{R}}(M, H_1, N, \mathrm{rank}(A_1^0 B_1^0)). \tag{61}$$

For $\overline{\lambda}_2$, the set of the zero point is $\{(A_2^0, B_2^0)\}$, i.e., the corresponding model is regular. Thus, $\overline{\lambda}_2$ is equal to a half of the dimension:

$$\overline{\lambda}_2 = \frac{H_2(M+N)}{2}. \tag{62}$$

Therefore, the RLCT with regard to $\overline{K}(A, B)$ is denoted by $\overline{\lambda_{\mathrm{P}}}$, and

$$\overline{\lambda_{\mathrm{P}}} = \lambda_{\mathrm{R}}(M, H_1, N, \mathrm{rank}(A_1^0 B_1^0)) + \frac{H_2(M+N)}{2} \tag{63}$$

holds. Because of $K(A, B) \leqq \overline{K}(A, B)$, i.e., a $\lambda_{\mathrm{P}} \leqq \overline{\lambda_{\mathrm{P}}}$, we obtain the main theorem:

$$\lambda_{\mathrm{P}} \leqq \lambda_{\mathrm{R}}(M, H_1, N, r') + \frac{H_2(M+N)}{2}. \tag{64}$$

$\square$

