# OpenReview forum: "Upper Bound of Bayesian Generalization Error in Partial Concept Bottleneck Model"
_TMLR — Rejected by TMLR_

### Review · Reviewer_da3J · 2024-03-19

**Summary Of Contributions:**

To make neural networks interpretable, one strategy is to have in the last layer before the output a limited number of nodes which one hopes will represent concepts important for the output.  This strategy therefore uses "concept bottleneck models" (CBM).  Applying this strategy however, may decrease generalization error.  Therefore, a hybrid approach has been proposed, partial CBM (PCBM), which doesn't force all flow from internal nodes to output go through concept nodes.
The current paper now upper bounds the generalization error of PCBM, implying PCBM is usually preferrable over CBM.

**Audience:**

Yes

**Claims And Evidence:**

No

**Requested Changes:**

A first important step is to proofread the complete paper and polish the writing, not only for the few points I indicated above.  Once the paper is readable and we can verify the results (which I believe are plausible) it is likely the "claims and evidence" question can be answered more positively.  While polishing the text, please make it self-contained so a reader without prior expert knowledge can verify the result in a reasonable amount of time.

**Strengths And Weaknesses:**

My main concern is the writing, which makes the paper hard to read and the formalization/explanation hard to follow.
The result seems to be straightforward, and (as far as I could follow the explanation) sound;

Below, I answer no to the question "claims and evidence", mainly because to poor writing doesn't allow me to fully verify the result.

## Detailed comments (only a small sample of issues in the writing):

### Abstract:
* "Concept Bottleneck Model (CBM) is a methods": "is" is singular while "methods" is plural
* "has not been yet clariﬁed since PCBM is singular statistical model." : incorrect grammar

### Section 1
* "Methods of artiﬁcial intelligence such as neural networks has" : "methods" is plural while "has" is singular
* "intelligence systems" -> intelligent systems
* "In these ﬁelds, the models cannot be black boxes,": In practice, models in these fields often are black boxes.  You probably mean "In these fields, there is an interest into models which are not black boxes".
* "It has been empirically showed" -> It has been shown empirically
* "neural networks are statistically singular in general" : Later, the text defines "singular", but never "statistically singular".
* "the Kullback-Leibler (KL) divergence between q and p:" : previously the text introduced the notation p, i.e., p is a model.  But the text didn't define yet q.
* "a singular model has a higher generalization performance compared to that of a regular model" : this isn't necessarily true in general, there are hypothesis classes parameterized by a parameter w such that $w\to f(\cdot,w)$ is injective with higher generalization error than some other classes for which the map $w\to f(\cdot,w)$ is not injective.  The fact that the map is injective alone doesn't necessarily cause this.
* "In the previous research" -> In previous research

### Section 3.1

* "of random variables of n" -> of n random variables
* Typically in Bayesian inference, the data is given as input (evidence), so while you could call them random variables their value is fixed by the input.
* "$X^n$ and $Y^n$ be a collection" -> $X^n$ and $Y^n$ be collections (they are 2, plural)
* The notation "X^n" isn't ideal as the superscript is ambiguous (could also mean exponentiation)
* "The function value of $(X_i,Y_i)$" : $(X_i,Y_i)$ is a pair of random variables, not a function.  -> "The value of $(X_i,Y_i)$ ..."
* "where $\mathcal{X}$ and $\mathcal{Y}$ are subsets of a finite-dimensional space" : in practice often $\mathcal{X}$ and $\mathcal{Y}$ are subsets of different spaces.
* "real Euclidian or discrete space" : discrete spaces are not real, so you mean "real Euclidian space ordiscrete space".  usually Euclidian spaces are real, so it may not be necessary to use the word "real" here.
* Here $q$ is defined, which was already used in Section 1 (see above).
*  "the data-generating distribution is an induced measure of measurable functions $D_n$" : $D_n$ is a single dataset, which is inconsistent with the plural of "functions".  $D_n$ is a dataset, not a measurable function.  I don't understand what is a "measure of measurable function".
* Before Eq (2): "the function of $\mathcal{W}$, given as" -> "the function of $\mathcal{W}$ given as"
* "The free energy appears as a leading term in the diﬀerence between the data-generating distribution and model" -> you probably mean "... in the difference in log-likelihood between the data generating ...."
* Please clarify the expression $D_{KL}(q(Y^n|X^n) \| Z_n)$.  Here, both $q(Y^n|X^n)$ and $Z_n$ are real numbers, while $D_{KL}(\cdot, \cdot)$ typically operates on distributions rather than real numbers.
* In the expression $\mathbb{E}[F_n]$, over what do you take the expectation?
* "Evaluating the dissimilarity between the true and the predicted value is also important for statistics and machine learning. This perspective is called prediction." : do you mean "performance" ?
* "A predictive distribution is deﬁned by the following density function of a new output $y\in \mathcal{Y}$ with a new input $x\in\mathcal{X}$" -> do you mean " ...  output $y\in \mathcal{Y}$ given a new input $x\in\mathcal{X}$" ?
* "the Bayesian inference is deﬁned by inferring" : typically definitions don't involve inference.
* "data-generating distribution may be predictive" : what do you mean exactly?  do you want the data-generating distribution q to have the same form as $p^*$ in Eq (4) ?
* "A Bayesian generalization error $G_n$ is defined " -> One possible error measure for Bayesian generalization is G_n, defined ... "
* "Both of these perspectives ..." : I saw a "perspective" called "prediction" (maybe "performance" ???), but what is the second "perspective"?

### Section 3.2

* "As technical assumptions, we suppose the parameter set $\mathcal{W}$ is sufficiently wide" : please define "wide set"
* Definition 3.1: This is a first point where theory starts to deviate significantly from what is widely known in the field of machine learning.  At the same time, the text suddenly gets very brief, not providing definitions which would be needed to make the paper self-contained.  Please provide minimal information to follow the paper with basic prior knowledge, or if that is infeasible at least cite a paper which the details are provided.
* It seems surprising that the poles of $\zeta(z)$ are negative rational numbers, I looked up some of the provided references, but Watanabe 2018 is a complete book and reading a complete book is outside the available time for this review.

In Section 3.1, it seems the text covers quite basic notions from Bayesian learning theory, but presented in an insufficiently precise / rigorous way.  I understand mostly what is meant because I already know the theory.  As the text goes on, the same density of minor mistakes remains, making it increasingly difficult to follow where the text starts with the original contribution.  In order to be able to verify the result, it seems necessary to first rigorously polish the text.

---

> ### Author Response · Authors · 2024-04-25
>
> Dear Reviewer da3J,
>
> Thank you for your careful reading and comments.
>
> We reply to the comments as follows.
>
> ## Writing:
>
> Before submitting the original paper, we asked a company that specializes in correction to rewrite this paper and it was rewritten by a native English speaker.
> According to the above rewriting result, we corrected. grammar mistakes and typos.
> In this revision, we fix such errors pointed out in the review.
>
> ## Self-containedness:
>
> In order to make singular learning theory self-contained, we need a huge amount of explanation, and it would be difficult to explain them in a single paper, which would fill a complete book. We have provided the necessary explanation.
> According to your suggestion, we add an explanation how to calculate an RLCT when the normal crossing form is obtained by using resolution of singularities. We also cite reference of mathematics which are related to the fact that the zeta function can be analytically continued to a meromorphic function whose poles are negative rational numbers.

---

> > ### Comment · Reviewer_da3J · 2024-04-25
> >
> > Thanks for your reply.
> >
> > Writing: please note that companies with native English writers typically don't have the technical knowledge to ensure precise formulations of (technical) statements.  While I guess collaboration with such company may be beneficial, it is important that the result is not only featuring good English but also precise formulations and a rigorous technical development.
> >
> > With "self-contained", I mean that a paper ideally provides the definitions needed to understand the paper.  This should not take a full book if you restrict your preliminaries / definitions / existing results to what you need for your contributions.  I can imagine it takes a bit of space, if it would make it hard to fit the text in the recommended page limit, you are of course free to use an appendix for the more detailed explanations.  I guess most papers don't require more than 4-6 pages to carefully summarize what is needed before the reader starts digesting the novel contribution.
> >
> > You mention doing the changes mentioned in my review.  Please note that I said in my review that I only provided a "sample" of the issues.  If I would go systematically through the paper and note down all similar issues it would be my review going over the page limit.  The goal is that you carefully also (re-)proofread the rest of the paper and correct the (similar) issues which are there.

---

> ### Author Response · Authors · 2024-04-25
>
> ## Other Detailed Comments:
>
> > "In these ﬁelds, the models cannot be black boxes,": In practice, models in these fields often are black boxes. You probably mean "In these fields, there is an interest into models which are not black boxes".
>
> We fix the presentation subject to your suggestion.
>
> > "neural networks are statistically singular in general" : Later, the text defines "singular", but never "statistically singular".
>
> We fix it and consistently use "singular".
>
> > "the Kullback-Leibler (KL) divergence between q and p:" : previously the text introduced the notation p, i.e., p is a model. But the text didn't define yet q.
>
> We introduce the notation of q in the pointed text.
>
> > "a singular model has a higher generalization performance compared to that of a regular model" : this isn't necessarily true in general... The fact that the map is injective alone doesn't necessarily cause this.
>
> We fix the text and mention a hierarchical structure which is caused by the singularities (Note that this relationship is reciprocal, i.e., it can be said that singularities exist due to the hierarchical structure).
> This schematic written by an expert of singular learning theory is for your information:
> https://sites.google.com/view/sumiowatanabe/home/phase-transition-in-machine-learning
>
> > Typically in Bayesian inference, the data is given as input (evidence), so while you could call them random variables their value is fixed by the input.
>
> Singular learning theory treats the case that the data is random variables and the data-generating distribution is unknown. The fixed value is the realization of the random variable.
> Some information criteria such as WAIC and WBIC are based on this setting and widely used in many fields.
> We described this situation is considered as generic with some references in the first version.
> According to your pointing out we fix the text in the last paragraph of section 3.1 and specify the data is subject to a unknown distribution.
>
> > The notation "$X^n$" isn't ideal as the superscript is ambiguous (could also mean exponentiation).
>
> We fix them to $X^{(n)}$.
>
> > "The function value of " : is a pair of random variables, not a function. -> "The value of..."
>
> Random variables are measurable functions with regard to a probability measure.
>
> > "where and are subsets of a finite-dimensional space" : in practice often and are subsets of different spaces.
>
> We had a grammatical mistake. They are subsets of some spaces (not subsets of a fixed space). We fix the text.
>
> > "real Euclidian or discrete space" : discrete spaces are not real, so you mean "real Euclidian space ordiscrete space". usually Euclidian spaces are real, so it may not be necessary to use the word "real" here.
>
> We remove "real" from the text.
>
> > "the data-generating distribution is an induced measure of measurable functions" : is a single dataset, which is inconsistent with the plural of "functions".
> is a dataset, not a measurable function. I don't understand what is a "measure of measurable function".
>
> Since random variables are measurable functions, the collection of them induces a probability measure (distribution).
> The text provides a mathematical explanation of the data-generating distribution.
>
> > $D_{KL}(q(Y^n|X^n) \Vert Z_n)$
> > In the expression $\mathbb{E}[F_n]$, over what do you take the expectation?
>
> The expression $\mathbb{E}[]$ is an expectation operator on overall datasets.
> We fix the former KL divergence to $D_{KL}(Q \Vert Z_n)$, where $Q$ is the dataset-generating distribution.
> We also emphasize $Z_n$ is a distribution of the dataset in the revised text.
> Thank you for your pointing out.
>
> > "Evaluating the dissimilarity between the true and the predicted value is also important for statistics and machine learning. This perspective is called prediction." : do you mean "performance" ?
>
> No. We mean "prediction".
>
> > "data-generating distribution may be predictive" : what do you mean exactly? do you want the data-generating distribution q to have the same form as in Eq (4) ?
>
> Inferring that the data-generating distribution may be predictive is just an inference, not true.
> This sentence may be confusing. Hence, we remove it. Our study aims to analyzing generalization error rather than giving a methodology.
>
> > "A Bayesian generalization error is defined " -> One possible error measure for Bayesian generalization is $G_n$, defined ... "
>
> Bayesian generalization error is a technical term of singular learning theory.
>
> > "Both of these perspectives ..." : I saw a "perspective" called "prediction" (maybe "performance" ???), but what is the second "perspective"?
>
> Knowledge discovery and prediction. See line 3-4 in page 5 of the revised paper.
>
> > "As technical assumptions, we suppose the parameter set is sufficiently wide" : please define "wide set"
>
> It just means wide enough to include singularities. This adjective may be not needed. We fix the presentation.
>
> Thank you for your detailed review.

---

> > ### Comment · Reviewer_da3J · 2024-04-25
> >
> > Thanks for your answers.  I may check these issues when a review of a revision is needed.
> > Here I only provide some short notes on answers which look unsatisfactory at first sight:
> >
> > >>  "Evaluating the dissimilarity between the true and the predicted value is also important for statistics and machine learning. This perspective is called prediction." : do you mean "performance" ?
> >
> > > No. We mean "prediction".
> >
> > "Prediction" is the task of taking as input features and outputting an estimated target value.
> > Evaluating how much the predicted value differs from the true value is usually called "performance evaluation".
> > Maybe you also want to specify the performance criterion you use, e.g., "accuracy".
> >
> >
> > >>    "data-generating distribution may be predictive" : what do you mean exactly? do you want the data-generating distribution q to have the same form as in Eq (4) ?
> >
> > > Inferring that the data-generating distribution may be predictive is just an inference, not true. This sentence may be confusing. Hence, we remove it. Our study aims to analyzing generalization error rather than giving a methodology.
> >
> > I don't understand your answer as it is ungrammatical at several points (and probably also unclear, i.e., I can't guess a correction with correct grammar which would be a clear answer).
> >
> > >> "A Bayesian generalization error is defined " -> One possible error measure for Bayesian generalization is $G_n$ defined ... "
> >
> > > Bayesian generalization error is a technical term of singular learning theory.
> >
> > I'm not sure how your reply is an answer to my suggestion.

---

### Review · Reviewer_vfPa · 2024-04-09

**Summary Of Contributions:**

CBM assumes that concepts that are reasons for outputs are also observed, and these are inserted into the penultimate layer as observations. This is a method for explainability. Some data has missing concept observations, a problem which partial CBM attempts to solve. The generalisation performance of partial CBM has not been studied, despite empirical evidence that it may be as high as vanilla neural networks. The authors study Bayesian generalisation error of PCBM in three-layered linear models. This reveals that partial CBM decreases Bayesian generalisation error compared to CBM.

**Audience:**

Yes

**Broader Impact Concerns:**

I do not hold any immediate broader impact concerns.

**Claims And Evidence:**

Yes

**Requested Changes:**

- I believe there is a typo, or some kind of notationally inconsistency, in the paragraph under equation (3). The notation $D_{KL}(q(Y^n\mid X^n)\Vert Z_n)$ is used, however $Z_n$ is a normalising constant, not a distribution for which KL divergence is defined. By the way, how is KL divergence defined here (is singularity w.r.t. measures important, do we always use Lebesgue measure and not a discrete counting measure or other singular measure (e.g. spherical data))? E.g. in (6) and (7), what happens if the density/mass is zero at some point? What is the region of integration?
- My understanding of Theorem 3.2 is that it is a special calculation of the RLCT in the case of the two hidden layer reduced rank linear model in Definition 3.2. Is that correct? It would be nice to add in a sentence or two between Definition 3.2 and Theorem 3.2 to explain to the reader what the purpose of the theorem is.
- I am confused about two points in the equations (34) to (41). I think these are just presentation issues. I suggest trying to streamline the presentation of this series of equations
    - First, (40) exactly repeats (35) - (37). I am not sure why it is repeated.
    - Second, (41) exactly repeats (34) - (37). I am not sure why it is repeated.

Questions:
I found the discussion very interesting, but as this is not my area, some statements which may be trivial to experts were not obvious to me. It would be great if the authors could expand the discussion just a little bit.
- "These singularities cause that a singular model has a higher generalization performance compared to that of a regular model [refs]". Could you give some further intuition as to why singularities result in higher generalisation? Is it something to do with overparameterisation?
- "Almost all learning machines are singular [refs]." In what sense is the word "almost all" used --- is this a probabilistic statement? Would something like regression with the right matrix dimensions/SVM/PCA be included in this statement? Please give just one or two examples. Or is this what the next sentence is describing (The word "Other" is perhaps confusing me)?

Minor:
- Typo in abstract. "Concept Bottleneck Model (CBM) is a methods for" should be "Concept Bottleneck Model (CBM) is a method for".
- It could be nice to label Figure 1 according to the symbols $A$, $B$, $C$, $y$ and $x$ used in the text.
- Sometimes the term "generalisation error" is used, and sometimes "generalisation performance" is used. It might be better to use consistently only one, because, for example, it is unclear what "higher generalisation performance" means (I think this means "lower generalisation error"?)
- First sentence of section 3.1 does not make sense grammatically.
- The sentence under (13) doesn't make sense grammatically.
- Theorem 3.2 references Aoyagi, but no date.
- Six concepts are discussed in section 5. I would suggest labelling the start of each of these six points with \paragraph{}, to help the reader.

**Strengths And Weaknesses:**

Strengths:
- The paper is well-written, containing only minor grammatical issues. The problem to be solved is mostly clear, even for a non-expert. Lots of references are provided.
- The analysis relates to three-layer (two hidden layer) linear networks, and therefore a kind of low rank factorisation of linear regression. Even though the model is limited, this is still an important step.


Weaknesses:
- There are some undefined notations and functional signatures, particularly surrounding KL divergences.
- Theorem 4.2 appears to relate to Gaussian conditional distributions of the label and conditioning information given the input. In the commonly studied linear regression model, it makes sense for y to be Gaussian given x. I am not sure what it means for y and c to be jointly Gaussian given x. Perhaps the authors could elaborate on what this model means.
- No empirical results are provided.

---

> ### Author Response · Authors · 2024-04-25
>
> Dear Reviewer vfPa,
>
> Thank you for your careful reading and comments.
>
> We reply to the comments as follows.
>
> ## Requested Changes:
>
> > ... in the paragraph under equation (3). The notation $D_{KL}(q(Y^n|X^n) \Vert Z_n)$ is used, however $Z_n$ is a normalising constant, not a distribution for which KL divergence is defined.
>
> We fix the text and emphasize that $Z_n$ is a distribution of the dataset.
> Actually, $Z_n$ is a normalizing constant for a parameter $w$ but the integral in $Z_n$ removes the parameter and obtains such distribution just like marginalizing.
>
> > By the way, how is KL divergence defined here...
>
> It is defined as an integration of $y$. The integral range is the whole space. E.g. $Y$ is subject to a distribution whose support is $\mathbb{R}$, the integral range of $y$ is from the negative infinity to the positive infinity.
> We fix the former KL divergence to $D_{KL}(Q \Vert Z_n)$, where $Q$ is the dataset-generating distribution.
>
> > My understanding of Theorem 3.2 is that it is a special calculation of the RLCT in the case of the two hidden layer reduced rank linear model in Definition 3.2. Is that correct?
>
> Yes. Theorem 3.2. is a concreate result which is related to our result.
>
> > It would be nice to add in a sentence or two between Definition 3.2 and Theorem 3.2 to explain to the reader what the purpose of the theorem is.
>
> We add a sentence to explain what the theorem is.
>
> > I am confused about two points in the equations (34) to (41). I think these are just presentation issues. I suggest trying to streamline the presentation of this series of equations.
>
> The Eq. (40) in the first version ((46) in the revised version) shows that the RLCT of CBM is an addition of RLCTs of $H_1$-dimensional and $H_2$-dimensional part of concept.
> Replacing the former part with the RLCT of reduced rank regression as stated in the paper corresponds to replacing the concept of that part from explicit to tacit, that is, using PCBM instead of CBM.
> According to your suggestion, we specify the purpose of this repeating (theoretical evaluation for efficiency of the structure of PCBM compared with that of CBM).
> The Eq. (41) in the first version may be a redundant repeat; thus, we remove it.
> Thank you for your pointing out.
>
> ## Questions:
>
> > I found the discussion very interesting, but as this is not my area, some statements which may be trivial to experts were not obvious to me. It would be great if the authors could expand the discussion just a little bit.
>
> Thank you for your interest to our study. Below, we answer your questions.
>
> > "These singularities cause that a singular model has a higher generalization performance compared to that of a regular model [refs]". Could you give some further intuition as to why singularities result in higher generalisation? Is it something to do with overparameterisation?
>
> This schematic written by an expert of singular learning theory is for your information:
> https://sites.google.com/view/sumiowatanabe/home/phase-transition-in-machine-learning
>
> It visualizes an intuition of a relationship between a hierarchical structure and singularities and introduces some mathematical facts.
>
> > "Almost all learning machines are singular [refs]." In what sense is the word "almost all" used ... Or is this what the next sentence is describing (The word "Other" is perhaps confusing me)?
>
> This "almost all" is not terms of probability.
> The next sentence in the paper is describing examples (such as LDA, NMF, and etc.).
> As you said, our "Other" caused some confusion.
> We fix the text from "Other instances of ..." to "Such instances of ...".
> Thank you for your pointing out.
>
> ## Minor:
>
> * Typo and grammatical mistakes are fixed.
> * We label Figure 1 according to the symbols.
> * We replace "generalization performance" with "generalization error" for consistency. We also correct from "higher" performance to "lower" error.
> * We fix to refer Theorem 3.2 with date (using \citet).
> * We label paragraphs in Discussion.
>
> Thank you for your suggestion.

---

### Review · Reviewer_SLtw · 2024-04-14

**Summary Of Contributions:**

The work claims to theoretically analyze RLCT (real log-canonical threshold Watanabe 2009, 2018) and derives an upper bound on the Bayesian generalization error in PCBM (partial concept bottleneck model Sawada and Nakamura 2022). The final conclusion is that due to the error is larger in CBM than in PCBM, then one generalizes better giving the same architecture.

**Audience:**

No

**Claims And Evidence:**

No

**Requested Changes:**

The manuscript would need extremely major changes to have a chance of getting an acceptance score from my side, honestly. In its current state and also given the red-flag issue exposed, I do think it is not yet ready for publication at TMLR. Of course, if my concerns previously added are addressed in a thorough way -- I'm glad to reconsider my opinion.

**Strengths And Weaknesses:**

**Strengths**: The work introduces in an extremely detailed way RLCT, rederives it and also builds a follow-up result for PCBMs. The contribution is clear, despite it's not very novel in my honest opinion. No experiments are performed and the definition of the neural network architecture is simple, but sometimes not well-explained or easy to understand if some reproducibility is required.

**General Weaknesses**: Basically, the paper overclaims a bit in my opinion, since it mixes three areas: i) CBM, ii) Bayesian inference/modeling + generalization error, and iii) RLCT. From these three, little focus or rigor is placed in i) and ii) in my opinion, which is not easy to follow or at least is not well-aligned with other previous works in these directions in the community. Apart from that, the contribution is not very significant, as it is deriving the bound for the PCBM case (for CBM there was already a result) for a particular architecture under strong assumptions. It is kind of difficult to see a broad impact of this result in the future in its current state, and one reader might have the feeling that the work is just a follow-up result of the previous RLCT developments on CBM. Btw, these ones are re-derived in the paper in a kind of unnecessary way (Theorem 3.1 and Theorem 3.2).

Additionally, I detected some grammar mistakes in the intro (1st paragraph) and abstract that would be convenient to fix. Moreover, some statements are kind of confusing as they mix different concepts in a way that is not explained and difficult to follow. Some examples:

- It is explained that CBM was developed with the aim of making the architecture interpretable, but it is not really explained why then the focus is placed on generalization.
- The background around the singular and regular models and how one generalizes better than the other is not surprising to me, and my feeling is that it's just re-explaining with extra math what is already known.
- PCBM is introduced to make the supervision cost of CBM cheaper, but then again the focus is placed in generalization, which is rather confusing to me.

**Red Flag for AC and TMLR Chairs**: The work basically overcites every type of work around RLCT and super-closely related topics. The submitted manuscript basically has +75 references in 11 pages. Of these, at least approximately 33% (~25 works) come from the same author. If we add two other researchers, basically an amount close to 50% of the 75 citations come from 3-4 authors. Very naively, I see an intention here, that I believe goes against the research ethical policy of a journal like TMLR and its standards. It is very well known that similar practices are conducted in low-quality journals and even predator-like publications. This point is particularly easy to observe, as I don't think the RLCT theory is yet as popular to be cited in this way.

---

> ### Author Response · Authors · 2024-04-25
>
> Dear Reviewer SLtw,
>
> Thank you for reading and suggestion.
>
> We reply to the comments as follows.
>
> ## Why generalization?
>
> PCBM and some variants (such as CBM-AUC) have been provided for not only decreasing the annotation cost but also improving the generalization performance.
> It is important to clarify an effect to the generalization error when an architecture is applied.
> In this paper, we analyze the Bayesian generalization error in PCBM and compare it with that of CBM,
> which reveals how the generalization error is affected by partializing the concept of CBM.
>
> By the way, recently, there are some experiments which show RLCTs contribute interpretability.
> According to your comment, we cite several references and briefly introduce the contribution of RLCTs in section 2
> in order to show a relationship between determination of RLCT and making interpretability.
>
> ## Significance
>
> According to the official concept of TMLR (https://jmlr.org/tmlr/),
> > TMLR emphasizes technical correctness over subjective significance,
> > to ensure that we facilitate scientific discourse on topics that may not
> > yet be accepted in mainstream venues but may be important in the future.
>
> Therefore, TMLR aims to foster discussion of non-mainstream topics while emphasizing technical correctness.
> Singular learning theory, which is the field of our research, is certainly not mainstream,
> but we theoretically show what kind of effect the partialization of the CBM concept represents in interpretable machine learning.
> We believe it is technically correct and worthy of discussion even if they are not mainstream.
>
> Besides, singular learning theory is not over math.
> Conventional statistical theory which treats the regular case could not have clarified the behavior of the generalization error and the marginal likelihood in singular models.
> Singular learning theory has provided significant methods for model selection, WAIC and WBIC, which are widely used in many fields.
>
> ## References
>
> We emphasize that these references have no intension for cheating.
> There is a group who have investigated singular learning theory over the past two decades.
> We cite their papers to show that there are many studies and singular learning theory has a long history enough to be introduced to TMLR community.
> What makes a few authors stand out in the references is simply the fact that they have made significant contributions and played central roles in singular learning theory.
> It is legitimate to make such an introduction from our side.
>
> According to your comment, we add some references related to recent studies of singular learning theory.
> Due to the difference in the length of history, there are fewer examples of singular learning theory for deep learning in recent years than for past shallow models described in Related Works of the first version.
> Of course, we are aware that there are various types of statistical learning theory, for example, VC theory and some regimes (mean field, NTK, Langevin dynamics, etc.) to analyze optimizers and loss landscapes for deep neural networks,
> but we do not think that it is appropriate to cite papers from other research groups conducting the other theory because our paper mainly deals with singular learning theory.
> It is also not appropriate to refer unrelated and papers in the field of interpretable machine learning.

---

### Decision · Action_Editor_mAR2 · 2024-06-07

**Recommendation:** Reject

**Comment:**

The main issue with this paper that it not clearly written.
I therefore think that "reject and resubmit" decision is appropriate.
Please note that according to TMLR's evaluation criteria, "papers that aren’t clearly written" may be rejected.

To accept this paper, significant improvements in writing are essential, including enhancing readability and the use of appropriate terms. Authors are expected to take this feedback seriously and make necessary improvements.

I have two major concerns regarding the writing of this paper.
The first one is that the authors provided only minimal responses to many of the suggestions from Reviewer da3J in their revision. As Reviewer da3J indicated, these suggestions are just representative examples, and it is necessary to reconsider the readability and appropriateness of individual terms throughout the entire paper.

The second concern is that the authors did not adequately respond to Reviewer SLtw's concerns about reference bias.
I do not believe the authors had any ill intent to deliberately skew the citations.
The perceived bias seems to stem from the fact that the topic is not widely studied and the community is small.
However, the authors should be aware that simply citing many papers from the same field in the Related Work section is not an ideal way.
The authors should focus on citing and discussing papers that are particularly relevant to the main subject of the paper.
It would be preferable to avoid citing papers that are not deeply related to the study, and instead, if possible, cite books or surveys to provide an overview of the entire field.
For example, in the case of this paper, the following books might serve as suitable references to provide an overview of the field.
* Sumio Watanabe. Algebraix Geometry and Statistical Learning Theory. Cambridge University Press, 2009.
* Sumio Watanabe. Mathematical theory of Bayesian statistics. CRC Press, 2018.

Below are the points I noticed while reading the paper. Please note that these are not the only points that need revision. I expect the authors to use appropriate terms, ensure the underlying messages of each sentences are properly conveyed to readers, and make the entire paper easier to read.

1. **[Abstract] "the Bayesian generalization error in PCBM"**: This will "of" instead of "in".

2. **[Page 1] "the concept bottleneck structure limits the parameter region of the network and it decreases the generalization error increases"**: What does "it decreases the generalization error increases" mean?

3. **[Page 1] Paragraph beginning "Sawada & Nakamura (2022) proposed ..."**: The main focus of this paper is the analysis of PCBM. Although CBM-AUC is a variant of PCBM, it is not the exact subject of analysis. Starting the second paragraph with CBM-AUC may mislead readers into thinking the main focus is on CBM-AUC. The topic sentence should convey the primary message of the paragraph.

4. **[Page 3] "there are a positive rational number λ and ..."**: It would be "there is".

5. **[Page 3] "Almost all learning machines are singular"**: It is unnatural to put this general statement after "mixture models are singular" and "neural networks are singular". If making such a strong, general claim, it should be stated at the beginning of the paragraph before mentioning specific cases.

6. **[Page 3] "Such instances of the singular learning theory applied for concrete models include the Boltzmann machines ..."**: What exactly does "Such instances" refer to here? How does this relate to the previous statement that "Almost all learning machines are singular"? Also, "including" would be appropriate.

7. **[Page 4] "The free energy appears as a leading term in the difference in log-likelihood between the data-generating distribution and model used for the dataset-generating process."**: What is the difference between "the data-generating distribution" and "model used for the dataset generating process"?

8. **[Page 5] "This perspective is called knowledge discovery."**: Is this term used in existing literature or newly introduced in this paper? If the former is the case, a reference is needed. Also, why is model selection (e.g., determining the number of hidden units) considered knowledge discovery? A brief explanation is needed.

9. **[Page 5] "This perspective is called prediction."**: As pointed out by Reviewer da3J, this statement is questionable. The preceding sentence, which is the paragraph's topic sentence, emphasizes "Evaluating the dissimilarity," so it is expected that a term related to evaluation follows, such as "performance." The authors' intent is not correctly conveyed here.

10. **[Page 5] "the data-generating distribution and predictive one"**: If referring to equation (4), "the data-generating distribution q and predictive one p^*" would be easier to understand.

11. **[Page 6] "Mathematical properties and details of ... We refer to the above complex function ζ(z) as the zeta function of learning theory"**: These two sentences should be reordered. Definitions of symbols and terms should be kept close together to avoid confusion.

12. **[Page 6] "RLCT of the model with regard to K(w)" and "the RLCT with regard to K(w)"**: In Definition 3.1 and equation (11), both are given the same symbol ζ(z). If they need to be treated as different ones, different symbols should be used. Also, which ζ(z) is the subject of analysis in this paper? If it is the latter, I think it would be approproiate to adopt the latter as Definition 3.1.

13. **[Page 6] "K(w0) = ∇K(w0) = 0"**: The first equation seems inappropriate because the dimensions do not match.

14. **[Page 6] Equations (16) - (18)**: The exponent $2kz + u^h$ should be $2kz + h$.

15. **[Page 7] Theorem 3.1**: A reference (Watanabe ...) should be added as in Theorem 3.2.

16. **[Page 7] Definition 3.2 "It is holomorphic in R(z) > 0 and can be analytically continued as a meromorphic function on the entire complex plain."**: If I understand correctly, this is not a definition but a property that follows from the definition of equation (24).

17. **[Page 8] "the input x is observable, w = (A, B) is a parameter and the output y and concept c is randomly generated"**: "w = (A, B) is a parameter, and the output y and concept c *are* randomly generated" would be more appropriate.

18. **[Page 8] Definition 4.1**: The latter part of this definition is not a definition but a property that follows from the definition. If so, it should be stated outside the definition, and proof may be necessary.

19. **[Page 8] "It is immediately derived that λP is a positive rational number."**: This does not seem immediate to me. If a simple proof suffices, it should be added to the appendix.

20. **[Page 9] "PCBM is an interpretable machine learning model; thus, it can be applied to not only prediction of unknown data but also explanation of phenomenon, i.e., knowledge discovery."**: Is "knowledge discovery" here referring to model selection mentioned in page 5? If so, can determining the number of hidden units in PCBM be considered an explanation of the phenomenon? If it is used in a different sense, a different term should be used.

21. **[Page 9] "Theorem 4.2 is immediately obtained"**: It would be more reader-friendly to add "from Theorem 3.1" or similar.

22. **[Page 9] "the Bayesian predictive distribution namely includes point estimation"**: The use of "namely" here is unusual.

23. **[Page 10] "To consider the effect of PCBM, i.e., partial replacing concepts"**: "partially" would be more appropriate.

24. **[Page 11] Corollary 5.1 "smaller for PCBM, which gives the observations for only the H2 dimension of the middle layer, than for CBM, which gives the observations for all of the middle layers"**: "PCBM that gives ...", "CBM that gives ..." is clearer.

25. **[Page 11] "our main result shows that PCBM is preferred to the upper model for generalization."**: It is not clear how this claim follows from the main result. More detailed explanation is needed.

26. **[Page 12] "Lastly, remaining problems are discussed"**: This sentence is redundant with the paragraph's title.

27. **[Page 12] "The other issues are as follows."**: In a paragraph dealing with "Remaining Problems," it is unclear why this sentence is needed between individual topics. Is there an intention to treat the topic before and after this sentence differently?

**Audience:**

This paper analyzes the Bayesian generalization error of the Partial Concept Bottleneck Model (PCBM), which falls within the scope of TMLR.
Analyzing the generalization error of deep neural networks, including PCBM, is a major topic in learning theory.

**Claims And Evidence:**

The main contribution of this paper is providing an upper bound on the Bayesian generalization error of the Partial Concept Bottleneck Model (PCBM). The paper includes a basic overview of singular learning theory and related studies, the main theorem concerning the real log canonical threshold (RLCT) of PCBM, and a discussion of the implications of these findings.
The paper's argument appears to be correct which extends Theorem 3.2 (Aoyagi & Watanabe (2005)) to PCBM.
However, as I note in Comment below, the paper needs significant improvement in terms of writing and the use of appropriate terms.

**Resubmission Of Major Revision:**

The authors may consider submitting a major revision at a later time.